# Access to Information and Communication Technology, Digital Skills, and Perceived Well-Being among Older Adults in Hong Kong

**DOI:** 10.3390/ijerph20136208

**Published:** 2023-06-23

**Authors:** Kwok-kin Fung, Shirley Suet-lin Hung, Daniel W. L. Lai, Michelle H. Y. Shum, Hong-wang Fung, Langjie He

**Affiliations:** 1Department of Social Work, Hong Kong Baptist University, Hong Kong, China; kkfung@hkbu.edu.hk (K.-k.F.);; 2Faculty of Social Sciences, Hong Kong Baptist University, Hong Kong, China

**Keywords:** internet access, digital divide, well-being, aging, public health

## Abstract

Population aging is a major concern worldwide. Active aging should be promoted by increasing the social participation of older adults and enabling them to remain involved in the community. Research has demonstrated the utility of digital resources for addressing the needs of older adults, which include networking, entertaining, and seeking health-related information. However, the digital divide among older adults (i.e., the “gray digital divide”) is increasingly being recognized as a social problem that may be related to poor well-being. To obtain updated local data on the prevalence of Internet access and usage and their relationship with perceived well-being, we conducted telephone interviews with a random sample of 1018 older adults in Hong Kong from January to July 2021 (This research has received funding support from the Interdisciplinary Research Matching Scheme, Hong Kong Baptist University). We found that only 76.5% of the participants had Internet access at home, a mobile phone data plan, or both, whereas 36.3% had never used Internet services and 18.2% had no digital devices. A younger age, male gender, higher education level, living with others, and higher self-perceived social class were associated with material access to digital devices and everyday use of Internet services. Participants who accessed the Internet every day had higher levels of life satisfaction and self-rated physical and mental health than those who rarely or never used the Internet. Hierarchical multiple regression analysis revealed that digital skills were significantly associated with self-rated mental health even when controlling for demographic variables (including age, gender, education level, and self-perceived social class). This study contributes to the limited body of literature on the relationship between Internet use, digital skills, and perceived well-being. Our findings highlight the importance of addressing the digital divide even in cities with high penetration of information and communication technology, such as Hong Kong. We also discuss our insights into the use of timely interventions for older adults to address the gray digital divide.

## 1. Background

Population aging is a major concern worldwide. The proportion of the world’s population older than 60 years is expected to nearly double from 2015 to 2050 [1]. This rapid population aging has been reported to occur concurrently with reductions in birth rates and increases in life expectancies [2]. Population aging is of concern because it lowers participation in the labor force and increases demand for medical and social care resources [3]. The association between older age and a smaller social network [4] implies that older adults may have less social capital than younger people, which could lead to poor well-being [5]. In particular, the social and physical environments of older adults changed considerably during the COVID-19 pandemic, as limitations were imposed on their mobility and social connections. Older adults have been threatened by the loss of social engagement [6] because of their vulnerability to infectious diseases and the harmful effects of social distancing restrictions during the pandemic [6]. The impacts of the COVID-19 pandemic on older adults have also involved barriers to accessing health and social services and even to continuing their daily and social activities [7]. The pandemic thus has magnified the significance of population aging as a major social concern [6].

Numerous studies conducted in response to population aging and its social and individual consequences have highlighted the significance of promoting active aging through, for example, increasing the social participation of older adults and enabling them to remain involved in the community [8]. In particular, it is important to create supportive environments for older adults to enable their active involvement in the community; thus, it is important for sufficient infrastructure and services to be available to meet their needs [9]. Developments in information and communication technology (ICT) have facilitated the belief that such tools can cater to the daily needs of older adults [10,11]. Currently, digitalized services cover various areas of life and help fulfill the needs of older adults such as acquiring information, including health-related information; staying connected with family and friends and participating in various types of e-shopping and e-banking [12]. The adoption and use of ICT can substantially counteract older adults’ increased risks of becoming homebound and isolated because of physiological deterioration that constrains mobility and diminishes social networks [13]. However, digital accessibility has excluded important social groups such as older adults [12]. Studies aiming to unravel this phenomenon have coined the term “digital divide,” which generally refers to “a division between people who have access to and use of digital media and those who do not” [14] (p. 2). Despite continuous developments in ICT in past decades, the digital divide has persisted and has been categorized into different levels [15,16]. The first-level digital divide focuses on the challenges associated with, or even lack of access to, physical digital devices and the Internet among certain social groups; the second-level digital divide refers to the inadequate grasp of digital knowledge and skills, i.e., digital literacy, of social groups that has constrained their access to ICT; additionally, the third-level digital divide refers to the differences in real-life benefits derived from digital access between different social groups [17,18]. Disadvantaged groups have been found to be prevalent among those suffering from the digital divide at different levels [15,16,19]. The relationship between social inequality and the digital divide at various levels has received increasing attention in recent years [18,20]. In particular, the term “gray digital divide” is used to refer to the digital divide that affects older adults [21]. In addition, older adults have been found to encounter more challenges when using ICT because of their low level or even lack of knowledge and digital skills or related disadvantages (i.e., the second-level digital divide) [22,23]. Because of these challenges, older adults may find it more difficult to rely on ICT to meet their individual needs (e.g., networking, looking for information) than younger people, even though older adults may especially need these technologies. Thus, older adults’ well-being might be affected by a lack of access to ICT and/or digital skills, which in turn affects the quality of digital access. Studies focused on the real-life benefits of digital access (i.e., the third-level digital divide) have shown that ICT use may be associated with improvements in various aspects of well-being among older adults [24,25]. In particular, one study found that digital skills mediated the relationship between online information-seeking and subjective well-being [26]. Another large-scale study revealed that higher levels of Internet use were associated with lower levels of loneliness and higher levels of social support, life satisfaction and mental well-being among older adults [27]. Despite these substantial benefits for older adults, other studies have revealed that older adults received relatively lower levels of benefits than the dominant social groups [19,28,29,30]. In addition, differences in the benefits attained within populations of older adults are currently among the most popular research topics [31,32]. Enabling older adults in general, and vulnerable older adults in particular, to attain these already low levels of real-life benefits relative to those accrued by dominant groups has attracted attention among researchers studying the third-level digital divide in recent years [33,34,35,36].

In light of the literature on this topic, digital inequality should be given greater consideration from the standpoint of public health, as it may be linked to a low level of well-being among older adults [37,38]. To assist future studies in this area and inform the creation of interventions to target the requirements of older adults, increased knowledge of the prevalence of digital access, the digital skills acquired and their relationships with ICT usage among older adults is essential. The current study examined these difficulties in a representative sample of older adults in Hong Kong while keeping these issues in mind. We were especially interested in comprehending and addressing the digital divide among Hong Kong’s older adults.

In Hong Kong, a large city with a distinctive culture, the gray digital divide is increasingly becoming a societal problem [2]. More importantly, Hong Kong is one of the most rapidly aging cities in the world. Its aging population increased from 13% of the total population in 2011 to 20% in 2021 [39]. Even though Hong Kong has excelled in the development of its digital infrastructure and has reported high rates of smartphone and Internet accessibility in recent years, the issue of digital exclusion among older adults remains critical.

According to the Thematic Household Survey Report No. 69, published by the Census and Statistics Department of the Hong Kong Special Administrative Region [40], Internet usage rates ranged from 97.5% to 99.9% among Hong Kong residents aged 10 to 64, compared with only 62.2% among residents aged 65 or older. Similarly, although the rate of Internet usage during the past 12 months more than doubled (from 24% to 62.2%) among Hong Kong residents aged 65 or older from 2015 to 2019, the rate of Internet penetration among older adults remains far behind that among young people (i.e., those aged 15 to 24) [40]. These figures clearly show that older adults in Hong Kong remain disadvantaged and marginalized in this international city with a high level of ICT penetration.

Additional research is needed to obtain information about ICT usage and digital skills among local older adults. The government data are not sufficiently comprehensive, and earlier research on the digital divide among older adults in Hong Kong [41] was based on secondary data and requires updating. More importantly, ICT usage may have changed over time, especially in the post-COVID-19 pandemic era, because of the unprecedented emphasis on ICT use. Some studies have suggested that a lack of sufficient knowledge and experience regarding the use of digital devices may explain the digital divide [2,41]. Besides the limitations in digital literacy and lack of friendly hardware designs (e.g., small screen sizes) faced by older adults [42], few studies have addressed the issues of ICT access and the digital divide among older adults in Hong Kong. Thus, a better understanding of this phenomenon is needed to inform the development of interventions to address the needs of these older adults.

### The Current Study

Against this background, the primary aim of this study was to investigate the prevalence of Internet access and use, digital skills and the perceived challenges associated with ICT use among older adults in Hong Kong.

In addition, we hypothesized that digital skills and digital use would be associated with better physical and mental well-being in our sample when we controlled for demographic variables. We based this hypothesis on the rationale that, as discussed above [27], Internet use may generate real-life benefits for older adults in terms of improved well-being. Although it is reasonable to assume that Internet use may promote well-being among older adults by satisfying various needs [43,44], a recent review of the limited literature on this topic did not find an association between Internet use and well-being [45]. Given the paucity of data on this topic and previous studies’ lack of consideration of digital skills, the current study also examined whether Internet use and digital skills are associated with physical and mental well-being.

## 2. Methods

### 2.1. Participants

Ethical approval for this study was obtained from the institutional review board at the Hong Kong Baptist University in Hong Kong. From January to July 2022, we conducted a telephone survey to investigate material access to ICT, the frequency of digital use, digital skills, and the self-rated well-being of older adults. Based on the total population of older adults aged 60 or above in Hong Kong (1,655,814), 1018 participants were randomly selected to achieve a confidence level of 95% with a margin of error of 3.1%. The dialed landline telephone numbers were randomly selected from a list of 8-digit Hong Kong household telephone numbers. The list consisted of known prefixes assigned to telecommunication service providers provided by the Office of the Communications Authority. Consent to participate in the study was attained from participants who were willing to disclose their age as 60 or older. The participants were divided by age into four cohorts, namely 60–69, 70–79, 80–89, and 90 and older which, respectively, included 404, 383, 187, and 44 participants. The telephone interviews were recorded and had an approximate duration of 15 min.

### 2.2. Measures

Each telephone interview included 45 questions to address the participant’s demographic background and apply the following measures.

*Access to the Internet and digital devices*. We asked the participants several questions about their access to digital devices and the Internet, including desktop, laptop, and tablet computers and smartphones [46]. The participants were asked to respond “yes” or “no” to indicate their access to these devices. We also asked the participants about where they accessed the Internet, e.g., at home or via mobile data outside the home. These items have been widely used in digital inclusion studies, and their validity has been established in the global context [46]. These items have also been used in the Chinese context [47,48].

*Internet use frequency*. We asked the participants about the extent to which they used the Internet for various activities, and responses were given on a 6-point scale (1 = Never, 6 = Every day). These Internet frequency items have been used in digital inclusion studies, and their validity has been established [49]. Again, these items have also been used in Chinese studies [47].

*Mobile Device Proficiency Questionnaire (MDPQ-28)*. This is a 28-item self-report measure of the respondents’ levels of different types of digital skills (e.g., data and file storage, communication, calendar use, and software management) [50]. The MDPQ-28 has been used in previous studies on digital inclusion issues [51]. The participants were asked to rate their grasp of skills using a 5-point scale (1 = Never; 2 = Very difficult; 3 = Difficult; 4 = Easy; 5 = Very easy). The psychometric properties of the MDPQ-28 have been established previously [51,52], and it has been used in the Chinese context too [53]. The Chinese version of the MDPQ-28 had excellent reliability (α = 0.951) in the present study; its psychometric properties will be reported elsewhere.

*Self-rated physical and mental well-being*. The interview included single-item measures of self-rated health (SRH) and self-rated mental health (SRMH); both measures have been widely used in population health studies, and their validity has been established [54,55]. The two items asked the participants to rate their physical and mental well-being, respectively, on a 5-point scale (1 = Very poor, 5 = Very good). Both measures have been used in previous studies in the Chinese context [56,57]. We also included a few study-specific items to better understand the participants’ perceived well-being (e.g., satisfaction with life, feeling empty, sense of happiness).

### 2.3. Data Analysis

SPSS 22.0 was used for the analysis. We conducted a descriptive analysis of the frequency of Internet access and use and examined the reliability of the MDPQ (digital skills) and the self-rated physical and mental well-being scales. We also examined the differences between the participants with respect to gender or the presence or absence of frequent ICT use via the independent sample t-test, one-way analysis of variance (ANOVA), and chi-squared test. We also conducted hierarchical multiple regression analyses to examine the relationships between Internet usage and digital skills with self-rated well-being while controlling for demographic background variables. Specifically, we controlled for the effects of variables such as age, gender, education level, and self-perceived social class.

## 3. Results

### 3.1. Sample Characteristics

A total of 1018 older adults provided informed consent and responded to the telephone survey. As 411 respondents refused to participate in the survey, the acceptance rate was 71.24%. Of the participants, 196 did not disclose their age; the ages of the other 822 participants ranged from 60 to 99 (mean [M] = 72.86; standard deviation [SD] = 8.74). The sample was skewed toward the younger cohorts (i.e., 60–69, 70–79). More than half of the participants were female (n = 660, 64.8%). Approximately 16.6% of the participants had a bachelor’s degree. A majority of the participants perceived themselves as grassroots (34.4%) or middle class (26.9%). Regarding their demographic backgrounds, gender-based differences were observed in education level (*p* < 0.001) and self-perceived class status (*p* = 0.032). The sample characteristics and gender differences are summarized in Appendix A.

### 3.2. Access to ICT and Frequency of Use

Statistics on the participants’ material access to ICT and frequency of access are summarized in Appendix A (also see Appendix A). The mean number of digital devices was 1.54 (SD = 1.20). A gender difference was observed, with men having more digital devices than women (M = 1.85, SD = 1.30 vs. M = 1.38, SD = 1.11), t(639.2) = 5.82, *p* < 0.001. In our sample, 18.2% of the participants reported that they had none of the digital devices mentioned in the interview (desktop, laptop, mobile phone, and tablet), and 70.5% of this subset was female. Among the participants who did not use digital devices, 31 were aged 60–69, 65 were aged 70–79, 65 were aged 80–89, and 22 were aged 90 or older; this latter group comprised half of all of the participants aged 90 or older.

In the study sample, 76.5% of the participants had Internet access at home, through a mobile phone data plan, or both. Female participants were more likely than male participants to report having none of these Internet access options (25.9% vs. 19.0%), *p* < 0.001. In the overall sample, as female participants were less likely to have Internet access than male participants, they were more likely to respond “never” to the question regarding the frequency of Internet access (40.8% vs. 28.2%) (*p* < 0.001). Interestingly, however, in the subsample of participants with Internet access (n = 648), female participants had a higher level of Internet use (*p* < 0.001), whereas male participants had a higher level of digital skills (*p* < 0.001).

In addition, participants who had a higher education level and lived with others were more likely than others to use the Internet every day, while participants who perceived themselves as grassroots were less likely than others to use the Internet every day (Appendix A).

### 3.3. Relationship between Internet Usage, Digital Skills and Well-Being

We compared the participants who never used the Internet with those who used the Internet rarely or every day via the chi-square test and one-way ANOVA (see Appendix A). Participants who used Internet services every day were more likely to be satisfied with their life (76.6% vs. 58.4% to 70.7%) and less likely to report a sense of emptiness (14.8% vs. 17.3% to 20.5%) than were those in the other two groups. In addition, participants who used Internet services had significantly higher SRH scores than those who never used the Internet, and those who used the Internet every day had significantly higher SRMH scores than those who rarely or never used the Internet (see Appendix A).

A hierarchical multiple regression analysis of the subsample of participants who had used the Internet and did not refuse to answer any demographic questions (n = 452) revealed that digital skills were a significant predictor (β = 0.159, *p* = 0.044) of SRH even when demographic variables were controlled (e.g., gender, education level, perceived social class, and age), but this variable did not result in a statistically significant increase in the prediction model (ΔF = 2.160, *p* = 0.117). Nevertheless, digital skills significantly predicted SRMH (β = 0.232, *p* = 0.003) even when demographic variables (including age) were controlled, and this variable led to a significant increase in R2 of 0.021, F = 8.231, *p* < 0.001, ΔF = 5.075, *p* = 0.007 (see Appendix A).

## 4. Discussion

This study contributes to the limited body of knowledge regarding the digital divide and its implications for the health of older adults in Hong Kong, an international city renowned for its high penetration of ICT. Specifically, we provide updated data on the prevalence of Internet access and use and digital skills collected from a random sample of older adults in Hong Kong; more importantly, we contribute to the growing literature on the complex relationship between Internet use and perceived well-being. This study has several major findings. (1) Relating to the prevalence of Internet access and usage, 76.5% of the participants had Internet access at home, via a mobile phone data plan, or both. (2) However, 18.2% of the participants had no access to digital devices, and a significant gender difference was observed in this subgroup. (3) A younger age, male gender, higher education level, living with others, and a higher self-perceived social class were associated with a higher frequency of daily Internet use (see Appendix A). (4) Digital skills were a significant predictor of SRMH even when demographic variables (including self-perceived social class and age) were controlled. These findings can considerably inform our understanding of the use of digital devices and related needs among older adults in urban societies with high levels of ICT penetration. The results may also inform the development of interventions to resolve the digital divide among older adults.

First, the digital divide has been categorized into three levels as discussedin the Introduction, andthe findings of the current study are particularly relevant to the first and third levels. Regarding the first-level digital divide, the current study reveals that the situation in Hong Kong resembles those of other cities with high ICT penetration rates where older adults face digital access challenges. Specifically, studies have unraveled the existence of even the first-level digital divide in affluent societies that have a well-developed Internet infrastructure and high Internet penetration rate [58,59]. In the case of Hong Kong, the rate of Internet use among residents aged 10 and older increased from 30.3% to 90.5% between 2000 and 2018 [40]. However, the findings of this study reveal that older adults have remained a disadvantaged group. The results show that only three-quarters of the participants reported having access to the Internet at home or elsewhere. Compared with other age groups, which have already attained the level up to 90.5% by 2018, the digital divide, in terms of access to digital devices and the Internet, was evident among older adults. There exists a substantial group of older adults, which is more than 20%, failing to obtain access to the Internet, even in recent years. In addition, this study revealed that access to digital devices and the Internet are associated with demographic variables such as age, gender, education level, and perceived social class status. Specifically, members of vulnerable groups such as women, older people, and those with a lower education level and a lower perceived social class status experienced lower levels of digital access. These findings are consistent with those of recently published research on the digital access of older adults in different parts of the world, although such research is limited [60]. According to the resources and appropriation theory [14,61], the personal (e.g., age and gender) and positional categorical inequalities (e.g., possession, education, economic status) that produce uneven distributions of social resources can facilitate inequality in access to ICT, with vulnerable groups experiencing challenges regarding digital access, thus sustaining the first-level digital divide. In other words, social inequality within a society plays a critical role. In addition, challenges in digital access can limit older adults’ attainment of real-life benefits such as retrieving health information, sustaining or extending their social networks, and participating in social and recreational activities; thus, these challenges can further reinforce their already vulnerable social positions or existing social inequality, as identified in the literature [14,62]. In particular, the digital divide has led to further inequality and social exclusion during the COVID-19 pandemic [38]. Therefore, it is imperative to improve access to ICT among older adults, and social inequality must be considered when addressing digital access in this population.

The findings of this study are also related to the third-level digital divide. We found that participants who used Internet services every day had higher levels of life satisfaction, SRH, and SRMH, which are health-related real-life benefits or concerns related to the third-level digital divide, than those who rarely or never used the Internet. Further analyses revealed that digital skills were significantly associated with SRMH even when we controlled for demographic variables (including education level, age, and self-perceived social class). In other words, participants who were frequent Internet users were more likely than other participants to be satisfied with their current life and to perceive themselves as healthy and less likely to feel empty. This finding implies that the mental well-being of older people might be improved by helping them to access the Internet and improve their digital skills, which would allow them to use the Internet to fulfill their psychosocial needs. Community service providers should consider providing additional educational programs to enable older people to learn how to use computers and smartphones. In the future, such programs should be evaluated to assess whether improvements in digital skills can improve the mental and social well-being of older people in a community. Furthermore, as revealed in Appendix A, digital usage and digital skills may vary by gender. This possibility should be taken into account when providing educational interventions for older people. For instance, women may require more support and time than men to master the skills needed to use a computer and smartphone.

The current study’s findings are in line with those of studies suggesting that the digital divide is a considerable problem associated with poor well-being among older adults [5,27]. This possibility is important, as older adults encounter many types of challenges, including decreased mobility due to physiological deterioration, diminished social networks, and consequent increases in social isolation. A recent systematic review showed that interventions intended to change digital behavior may be effective in improving the well-being of older adults [63]. Older adults who have no access to ICT or poor digital skills find it difficult to use ICT to address needs such as maintaining connections with family members and friends or seeking information related to health or other daily issues; therefore, they may face an increased risk of psychological or social problems and may feel empty or unsatisfied with their lives. Because of a lack of access to information and resources that can be easily accessed online (e.g., government information, social and medical resources, product or service options and prices, transport information, educational resources, and entertainment resources), these marginalized older adults may have unaddressed needs, leading to poor physical and mental well-being. Our findings, together with the literature, highlight the importance of improving older adults’ access to digital devices and the Internet and providing them with timely interventions or support to overcome the challenges of using online technology. For older adults, addressing the first-level digital divide may improve their health-related impact related to the third-level digital divide. Another important point is that, consistent with the recent literature [45], we found that digital skills were a significant predictor of SRH and SRMH, but Internet use was not. This finding implies that both improving older adults’ digital skills and providing them with access to ICT are important for improving their well-being by addressing the digital divide.

Although Hong Kong is an affluent society with a well-developed digital infrastructure, many older adults remain excluded and marginalized. This is crucial because the COVID-19 pandemic has rendered digital access a daily necessity rather than a luxury, especially for vulnerable groups such as older adults [64]. Echoing calls from recent studies related to the persistence of even the first-level digital divide in different parts of the world, this study indicates that it is particularly important to build supportive environments to address the digital divide among older adults in Hong Kong [65].

This study highlights the importance of addressing the digital divide even in a city with high ICT penetration, such as Hong Kong. Recently, the relationship between digital access and its health-related impacts, a concern pertaining to the third-level digital divide, has increased the significance of such an endeavor. This study further provides insights into the need to target social inequality and its potential implications for the digital divide among older adults when addressing the issue. Nevertheless, this study has several limitations. First, the survey was conducted via telephone interviews, which were limited to a duration of 15–20 min and therefore could not be sufficiently comprehensive. Second, only those with access to telephones have been included in the study. There is the possibility that the most vulnerable older adults who do not have access to landlines or smartphones could be excluded. However, this study also has three major strengths. First, we used a random sampling method to recruit a representative sample of local older adults. Second, the telephone survey in this study, despite its limitations, collected some data that were not self-reported. Third, by identifying health-related outcomes of the digital divide among older adults, we have advanced inquiries regarding the terrain of the third-level digital divide in Hong Kong and provided related evidence demonstrating that Internet access and use are correlated with physical and especially psychological health when demographic variables are controlled.

## 5. Conclusions

In this study, we conducted telephone interviews to investigate Internet access and use and its relationship with well-being in a representative sample of older adults in Hong Kong. We found that 36.3% of our participants had never used Internet services, and 18.2% had no digital devices. Despite the high level of ICT penetration in Hong Kong, this study found that many older adults in Hong Kong experienced a lack of digital access or had limited digital skills. The focus on first-level digital divide issues allowed some variables associated with Internet access to be identified and explored. In addition, we found that digital access and the level of digital skills were significantly associated with the participants’ SRMH even when we controlled for demographic variables. These findings related to the third-level digital divide highlight the importance of addressing the digital divide even in cities with a high level of ICT penetration such as Hong Kong. Despite the methodological limitations mentioned above, this study contributes to the limited body of literature on the relationship between Internet use, digital skills, and well-being. In view of the COVID-19 pandemic and the increasing significance of digital access for vulnerable groups, including older adults, we argue that the digital divide among older adults should be regarded as a significant social issue that warrants additional research and timely interventions. Social inequality and its potential implications regarding the digital divide among older adults are among the targets to be explored.

## Data Availability

Data sharing is unavailable due to reasons of privacy.

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
