# Peer review of "Access to Information and Communication Technology, Digital Skills, and Perceived Well-Being among Older Adults in Hong Kong"

_ijerph, 2023, doi:10.3390/ijerph20136208_

Round 1
Reviewer 1 Report
This study addresses the digital divide, a relevant topic in the context of progressive and challenging aging. I would like to point out some suggestions and questions.
Abstract:
The authors could include the date of the study (from --- to ---) in the abstract.
Introduction:
The authors described the objectives of the study in lines 90 and 119. To avoid repetition, I suggest summarizing these objectives in a single paragraph.
Methods
The authors could write an item about the survey: How long did the telephone survey last? (15-20 minutes). How many questions were included? Was the phone survey recorded?
Measures
To avoid repetitive writing, the authors could summarize on the use of the measurement items in previous studies [references], including in the Chinese context [references]
Results
-How many elderly people did not accept to participate in the survey?
-Little has been said about Table 3. I would like the authors to make a few comments about this table, as digital devices should be easy to use, especially in the context of the elderly.
Discussion
The first paragraph could include a summary of the work.
The major findings (The main findings (line 240) are interesting. I would like to understand if these results are similar to those in Europe, for example.
References
I recommend including some references from 2022 or 2023.
Author Response
Abstract:
The authors could include the date of the study (from --- to ---) in the abstract.
The date of study was added to the abstract. The study conducted telephone interviews with a random sample of 1,018 older adults in Hong Kong from Jan 2021 to July 2021.
Introduction:
The authors described the objectives of the study in lines 90 and 119. To avoid repetition, I suggest summarizing these objectives in a single paragraph.
The study's objectives were summarised in a single paragraph, lines 103-111.
Methods
The authors could write an item about the survey: How long did the telephone survey last? (15-20 minutes). How many questions were included? Was the phone survey recorded?
The telephone interview lasted 15 minutes (line 167-168) and consisted of 45 questions (line 170-171). The telephone survey was entirely recorded ((line 167-168).
The telephone survey was recorded and lasted around 15 minutes.
Measures
To avoid repetitive writing, the authors could summarize on the use of the measurement items in previous studies [references], including in the Chinese context [references]
Thank you very much for your comments. Please note that sample items were added in the measures section and that references from both original studies and local validation studies were cited.
For example, regarding the SRH and SRMH, reference 39 (Ahmad et al., 2014) is cited as it is a systematic review that summarizes the validity of SRMH, while reference 40 (Pérez-Zepeda et al., 2016) is an empirical study which investigated the validity of SRH. References 41 and 42 also provided evidence for the validity of SRMH and SRH, respectively, among Chinese-speaking populations. (Line 192-199)
One exception is the Mobile Device Proficiency Questionnaires (MDPQ-28). Only original studies were cited, but no studies on the psychometric properties of the Chinese version are available. Therefore, we stated that "the Chinese version of the MDPQ-28 had excellent reliability (α = .951) in the present study, while its psychometric properties will be reported elsewhere." (Line 183-191)
Results
-How many elderly people did not accept to participate in the survey?
411 respondents rejected to participate in the survey (line 215).
-Little has been said about Table 3. I would like the authors to make a few comments about this table, as digital devices should be easy to use, especially in the context of the elderly.
Thank you for pointing this out. Table 3 in fact shows that female participants were generally less confident in using digital technology. We have further discussed this finding and suggested that gender differences should be taken into account when teaching older people to use computer and smart phone (in the discussion section, Line 326-336).
Discussion
The first paragraph could include a summary of the work.
A paragraph for the summary of the work was placed at the beginning of discussion.
This study has several major findings. 1) Relating to the prevalence of Internet access and usage, 76.5% of the participants had Internet access at home, via a mobile phone data plan, or both. 2) However, 18.2% of the participants had no access to digital de-vices, and a significant gender difference was observed in this subgroup. 3) A younger age, male gender, higher education level, living with others, and a higher self-perceived social class were associated with a higher frequency of daily Internet use (see Table 4).
References
I recommend including some references from 2022 or 2023.
Marimuthu, R., Gupta, S., Stapleton, L., Duncan, D., & Pasik-Duncan, B. (2022). Challenging the Digital Divide: Factors Affecting the Availability, Adoption, and Acceptance of Future Technology in Elderly User Communities. Computer, 55(7), 56-66.
Liao, S. C., Chou, T. C., & Huang, C. H. (2022). Revisiting the development trajectory of the digital divide: A main path analysis approach. Technological Forecasting and Social Change, 179, 121607.
Goedhart, N. S., Verdonk, P., & Dedding, C. (2022). “Never good enough.” A situated understanding of the impact of digitalization on citizens living in a low socioeconomic position. Policy & Internet.1-21
Nguyen, M. H., Hunsaker, A., & Hargittai, E. (2022). Older adults’ online social engagement and social capital: The moderating role of Internet skills. Information, Communication & Society, 25(7), 942-958.
Tőkés, G. E. (2022). The Third-level Digital Divide among Elderly Hungarians in Romania. Acta Ethnographica Hungarica, 66(1), 241-259.

Reviewer 2 Report
The article explores access to information and communication technology among older adults.
The abstract of the paper is quite long. The sample in the abstract is 1,000; in the paper, it is 1018.
In general lines, the introduction does not provide relevant information about similar researches on the topic.
The objectives of the paper and hypothesis should be more clearly exposed
The paper is not contextually situated.
Lines 90: Collection data: ‘representative sample’ is largely considered a pleonasm since the sample is a priori representative of the target population.
The authors need to give more explanations on the following:
- prevalence of Internet access and use (indicator and measures)
- older adults
- ethical considerations are not sufficiently explained (lines 132)
- more data on the sample and random sample procedures (telephone survey). Why the authors considered that a 1,018 sample is ‘representative’ of the older adult population in Hong Kong? How was this random selection achieved? As well, explain: Margin of error is 3.1?
- what does mean digital inclusion studies and what does the Chinese context?
- measurements on Likert Scale regarding Internet use frequency: (1- never and 5- every day). Which are the scales 2,3, 4, since 1 is considered never and 5- every day?
Lines 157: confusing. What kind of validity? Statistical validity?
Is the test called the Mobile Device Proficiency Questionnaires (MDPQ-28) validated? Which was the original study? Reference 37 is mentioned in the paper. Unfortunately, reference 37 is a systematic review and not the prime author. Usually, psychometric tests are validated and have validity and consistency (proved and demonstrated). No references or studies to prove that Cronbach alfa is high (lines 166).
The Self-rated physical and mental well-being. Idem: the question of validity for the psychometric tests. Only two references were added, and one study is a systematic review.
Line 192: confusing.
Line 194: What does mean self-perceived class status?
Why the authors used hierarchical multiple regression? The authors need a theoretical and rational basis for testing a set of nesting models. What are independent and dependent variables? Do the variables are interval data? Do the dependent and predictor variables are considered interval variables?
Why F- statistic is written here in ΔF = 4.062? What is the value of F?
To see an increase in the model’s predictive power, what is the value of F for the models considered in the analysis? Which is the R - change increment? Mode summary, ANOVA and coefficient syntax are not properly provided.
The discussion section is not developed on hypothesis testing. The relationship between the digital divide and poor well-being is not demonstrated. The prevalence of internet access is not sufficiently developed in the paper.
The section method (or material and methods section) provides information about data collection and methods applied.
The results do not encapsulate the findings discovered, and the discussion does not reveal the implications of the findings.
The conclusions section could be extended to support the results, as well as to the limitations of the study.
Author Response
In general lines, the introduction does not provide relevant information about similar researches on the topic.
The sample size of the study was 1018. Sorry for confusion.
The objectives of the paper and hypothesis should be more clearly exposed
Thank you very much for the comments. We have revised the last section of the introduction and have now explicitly stated the objectives and hypotheses of the present study.
The paper is not contextually situated.
The studies on the digital divide among elderly adults were underdeveloped. The contextualization of the study was presented in the last three paragraphs of the Background section.
Lines 90: Collection data: ‘representative sample’ is largely considered a pleonasm since the sample is a priori representative of the target population.
The suggested change has been implemented through changing the phrase, ‘representative sample’ to ‘sample’.
The authors need to give more explanations on the following:
- prevalence of Internet access and use (indicator and measures)
The paragraph has been rewritten and references added to illustrate the meanings of prevalence of internet access and use.
- ethical considerations are not sufficiently explained (lines 132)
The sentence has been rewritten and the need to follow the procedure as well as the name of the committee, the Research Ethics Committee, has been added.
- more data on the sample and random sample procedures (telephone survey). Why the authors considered that a 1,018 sample is ‘representative’ of the older adult population in Hong Kong? How was this random selection achieved? As well, explain: Margin of error is 3.1?
We claimed that this was a representative sample of older adults in Hong Kong because we employed random sampling in this study.
- what does mean digital inclusion studies and what does the Chinese context?
The phrase digital divide/inclusion studies has been added to illustrate to readers that these studies are those relevant to the concerns of the paper, digital divide.
Concerning Chinese context, a phrase, “which requires proper translation of the related questions to Chinese and adaption to the local situations”, has been added to illustrate its meanings.
- measurements on Likert Scale regarding Internet use frequency: (1- never and 5- every day). Which are the scales 2,3, 4, since 1 is considered never and 5- every day?
Sorry for confusion. Please kindly note that the internet use frequency was coded from 1 to 6, 1 was coded as “never”, 2 was coded as “less than once a month”, 3 was coded as “once a month”, 4 was coded as “once a week”, 5 was coded as “everyday”, 6 was coded as "several times per day”.
Lines 157: confusing. What kind of validity? Statistical validity?
The meaning of validity has been clarified through adding phrase, “as proper measurement of the variable”. The added phrases have been shaded in green for reviewer’s easy identification.
Is the test called the Mobile Device Proficiency Questionnaires (MDPQ-28) validated? Which was the original study? Reference 37 is mentioned in the paper. Unfortunately, reference 37 is a systematic review and not the prime author. Usually, psychometric tests are validated and have validity and consistency (proved and demonstrated). No references or studies to prove that Cronbach alfa is high (lines 166).
Reference 35 which has been cited in the paper is the original study. As the related paragraph failed to express properly, the original study was thus identified explicitly, by adding the phrase, “developed by Rogue & Boot (2018)” to the relevant sentence. About examples of studies adopting the scale, it is the mistake of the authors in failing to put reference 36, which is an empirical study, into the relevant citation box as well. For reference 37, even though it is a systematic review, the authors have included only ‘validated measurements’ into their studies. Thus, reference 37 was included as support for the argument. To rectify the mistakes, reference 36 has been included, together with reference 37 as supports for the argument. The change has been shaded in green.
Concerning the reliability of the scale, the original phrase has failed to communicate the meaning properly, so it has been redrafted as follows:
“For the present study, the Chinese version of the MDPQ-28 had attained acceptable reliability (α = .951)”, and shaded in green for easy perusal.
The Self-rated physical and mental well-being. Idem: the question of validity for the psychometric tests. Only two references were added, and one study is a systematic review.
Please note that the cited articles indeed provided psychometric data regarding the SRH and SRMH in the literature. In particular, reference 39 (Ahmad et al., 2014) is a systematic review that summarizes the validity of SRMH, while reference 40 (Pérez-Zepeda et al., 2016) is an empirical study which investigated the validity of SRH. References 41 and 42 also provided evidence for the validity of SRMH and SRH, respectively, among Chinese-speaking populations.
Line 192: confusing.
The sentence has been redrafted to indicate explicitly the relatively inferior self-perception of the participants in terms of their self-perceived class status:
“A majority of participants (82%) perceived themselves as either grassroots/lowest (34.4%), lower middle class (20.8%), or middle class (26.9%). Their self-perceived class status was relatively low, which fell within grassroots/lowest to middle class only.”
Line 194: What does mean self-perceived class status?
Self-perceived social class refers to an individual's perception or subjective assessment of their own social status or class in society.
Why the authors used hierarchical multiple regression? The authors need a theoretical and rational basis for testing a set of nesting models. What are independent and dependent variables? Do the variables are interval data? Do the dependent and predictor variables are considered interval variables?
Thank you very much for the comments. Hierarchical multiple regression was conducted because we needed to investigate the association between digital skills and self-rated health and self-rated mental health while statistically controlling for the effects of demographic variables, including age, gender, education level, and self-perceived social class. Our hypotheses have been explicitly cited in the last part of the introduction section. In addition, we treated self-rated health and self-rated mental health as interval data (1 = very poor ; 5 = very good).
Why F- statistic is written here in ΔF = 4.062? What is the value of F?
Please note that we have reported the values of ΔR2 and ΔF along with other necessary figures in Table 5. These figures are required to show the statistically significant change from Model 1 to Model 2 in the regression analyses.
To see an increase in the model’s predictive power, what is the value of F for the models considered in the analysis? Which is the R - change increment? Mode summary, ANOVA and coefficient syntax are not properly provided.
Thank you for your comments. Please note that we have reported the values of ΔR2 and ΔF along with other necessary figures in Table 5. These figures are required to show the statistically significant change from Model 1 to Model 2 in the regression analyses.
The discussion section is not developed on hypothesis testing. The relationship between the digital divide and poor well-being is not demonstrated. The prevalence of internet access is not sufficiently developed in the paper.
Thank you again for the suggestion. To strengthen the relationship between digital divide and poor well-being as well as the implications relating to the (insufficient) prevalence of internet access for the older adults, different related parts in the Background and the Discussion section of the paper has been redrafted (and shaded in green within the paper) as follows:
For the background:
… Despite continuous developments in ICT in past decades, the digital divide has per-sisted and has been categorized into different levels [15, 16]. The first-level digital di-vide focuses on the challenges associated with, or even lack of access to, physical digi-tal devices and the Internet among certain social groups; the second-level digital di-vide refers to the inadequate grasp of digital knowledge and skills, i.e., digital literacy, of social groups that has constrained their access to ICT; and the third-level digital di-vide refers to the differences in real-life benefits derived from digital access between different social groups [17, 18]. Disadvantaged groups have been found to be prevalent among those suffering from the digital divide at different levels [15, 16, 19]. The rela-tionship between social inequality and the digital divide at various levels has received increasing attention in recent years [18, 20]. In particular, the term “grey digital di-vide” is used to refer to the digital divide that affects older adults [21].]…
… Thus, their well-being might be affected as a result of a lack of access to ICT and/or digital skills which affects the resultant qualities of their digital access. Focusing on the real life benefits of digital access or third level digital divide, previous studies have shown that ICT use may be associated with better well-being for older adults in different aspects [18,19].
For the discussion:
… 1) Relating to the prevalence of Internet access and usage, 76.5% of participants in our sample had either or both Internet access at home and mobile phone data plan;…
… ]. However, this study showed that, of the prevalence of Internet access and usage, only three quarters of older adults reported having access to Internet at home or outside. Comparing to other age groups, which has already attained the level up to 90.5% by 2018, the digital divide for older adults, in terms of access to digital device and internet, was evident. There exists a substantial group of older adults, which is more than 20%, failing to get access to Internet, even in recent years….
…. This finding implies that one way to improve the mental well-being of older people may be to enable them to access Internet and improve their digital skills so that they could use Internet as a means to fulfil their psychosocial needs. Community service providers should consider providing more educational programs for older people to learn about how to use computers and smart phones. Such programs should also be evaluated in the future to assess whether the improvements of digital skills could in turn improve the mental and social well-being of older people in the community. Furthermore, as revealed in Table 3, there may be gender differences in digital usage and digital skills. Therefore, this should be taken into account when providing educational interventions for older people. For instance, females may require more support and time to master the skills to use computer and smart phone.
The section method (or material and methods section) provides information about data collection and methods applied.
The results do not encapsulate the findings discovered, and the discussion does not reveal the implications of the findings.
Thank you very much for reminding us to further discuss the implications of the findings. We have now added a paragraph to discuss the potential importance of providing digital skills training for older people to improve their mental and social well-being (in the discussion section).
The conclusions section could be extended to support the results, as well as to the limitations of the study.
Thank you for your comments. Please note that the limitations of this study have been acknowledged in the last part of the discussion section. We have also edited the conclusion sections to clarify our findings and the conclusions based on our findings.

Reviewer 3 Report
General:
· Major grammar/spelling checks throughout (e.g. consistent spelling of aging, major grammar issues)
· Refrain from using the term “elderly” as it promotes negative stereotypes (there are 2 uses of this word within the manuscript). Instead, use terms such as older adults.
Introduction:
· Authors talk about the three distinct levels of the digital divide extensively within the discussion, and, although the digital divide is mentioned within the introduction, the distinct levels are not. Authors should include discussion/or at least the relevance of the digital divide levels within the introduction and why the levels are important.
· Authors mention social inequality and its connection to digital access within the discussion, but do not mention this within the introduction. Authors should touch on this topic within the introduction.
Methods:
· Was the phone number list inclusive of those with both landlines and cell phones? Were you stratifying by age as well – to get an even distribution among “young-old” through “oldest-old?” Younger older adults (aged 75 and younger) might be more likely to be aware of internet and other computer issues as opposed to the oldest old (75-84, or 85 and older).
· Line 162-263 – The scale used to rate their grasp of skills ranged from 1, which was “never” and 5, which was “very easy.” What was the specific question, if “never” was a potential response? I would assume the scale would be 1-5 with one being “not at all easy” and 5 being “very easy.”
Results:
· Line 189 - 196 individuals did not disclose their ages, are researchers certain their age was a minimum of 60? Also, did the sample skew younger, or older?
· Line 192 - Can the authors define “grassroots” and “middle-class?”
· Line 201-202 – 70.5% of individuals without any digital devices were female. What was the age distribution of those without any digital devices? Were females more likely to be older? Could this be an age issue as opposed to a gendered issue?
· Line 225 – Was age also controlled for within the hierarchical multiple regression? If not, why? Again, “younger older adults” may be more likely to use technology than “the oldest older adults.” I can see age being an important predictor.
Limitation:
· One additional limitation which should be included is that only those individuals with access to telephones were included in the study, which excluded the most vulnerable older adults who do not have access to either a landline or smart phone.
Author Response
General:
- Major grammar/spelling checks throughout (e.g. consistent spelling of aging, major grammar issues)
Thank you for the suggestions. The paper has been subjected to grammar and spelling checks to deal with the identified issues.
- Refrain from using the term “elderly” as it promotes negative stereotypes (there are 2 uses of this word within the manuscript). Instead, use terms such as older adults.
The term “elderly” was replaced by “older adults” in this article
Introduction:
- Authors talk about the three distinct levels of the digital divide extensively within the discussion, and, although the digital divide is mentioned within the introduction, the distinct levels are not. Authors should include discussion/or at least the relevance of the digital divide levels within the introduction and why the levels are important.
- Authors mention social inequality and its connection to digital access within the discussion, but do not mention this within the introduction. Authors should touch on this topic within the introduction.
Responding to the precious advice from different reviewers, the background has the relevant parts redrafted and shaded in green in the paper for reviewer’s perusal:
Despite continuous developments in ICT in past decades, the digital divide has persisted and has been categorized into different levels [15, 16]. The first-level digital divide focuses on the challenges associated with, or even lack of access to, physical digital devices and the Internet among certain social groups; the second-level digital divide refers to the inadequate grasp of digital knowledge and skills, i.e., digital literacy, of social groups that has constrained their access to ICT; and the third-level digital divide refers to the differences in real-life benefits derived from digital access between different social groups [17, 18]. Disadvantaged groups have been found to be prevalent among those suffering from the digital divide at different levels [15, 16, 19]. The relationship between social inequality and the digital divide at various levels has received increasing attention in recent years [18, 20]. In particular, the term “gray digital divide” is used to refer to the digital divide that affects older adults [21].
… Thus, older adults’ well-being might be affected by a lack of access to ICT and/or digital skills, which in turn affect the quality of digital access. Studies focused on the real-life benefits of digital access (i.e., the third-level digital divide) have shown that ICT use may be associated with improvements in various aspects of well-being among older adults [24, 25].
Despite these substantial benefits for older adults, other studies have revealed that older adults received relatively lower levels of benefits than the dominant social groups [19; 28; 29; 30]. In addition, differences in the benefits attained within populations of older adults are currently among the most popular research topics [31; 32]. Enabling older adults in general, and vulnerable older adults in particular, to attain these already low levels of real-life benefits relative to those accrued by dominant groups has attracted attention among researchers studying the third-level digital divide in recent years [33; 34; 35; 36].
In light of the literature on this topic, digital inequality should be given greater consideration from the standpoint of public health, as it may be linked to a low level of well-being among older adults [37,38]. To assist future studies in this area and influence the creation of interventions to target the requirements of older adults, increased knowledge of the prevalence of digital access, the digital skills acquired and their relationships with ICT usage among older adults is essential. The current study examined these difficulties in a representative sample of older adults in Hong Kong while keeping these issues in mind. We were especially interested in comprehending and addressing the digital divide among Hong Kong’s older adults.
Methods:
- Was the phone number list inclusive of those with both landlines and cell phones? Were you stratifying by age as well – to get an even distribution among “young-old” through “oldest-old?” Younger older adults (aged 75 and younger) might be more likely to be aware of internet and other computer issues as opposed to the oldest old (75-84, or 85 and older).
The phone number list only included landlines. The age was divided into four age groups: 60-69, 70-79, 80-89, and 90 and older. These four age cohorts were distributed as follows: 404, 383, 187, and 44.
- Line 162-263 – The scale used to rate their grasp of skills ranged from 1, which was “never” and 5, which was “very easy.” What was the specific question, if “never” was a potential response? I would assume the scale would be 1-5 with one being “not at all easy” and 5 being “very easy.”
Thank you very much for pointing this out. Please note that we adapted the scale from previous projects as cited in the measures section (e.g., 35, 36). This scale asked about the level of difficulty experienced by participants when using mobile devices for various functions. The scale provides the following response options: 1 = Never; 2 = Very difficult; 3 = Difficult; 4 = Easy; 5 = Very easy.
Results:
- Line 189 - 196 individuals did not disclose their ages, are researchers certain their age was a minimum of 60? Also, did the sample skew younger, or older?
At the start of the telephone survey, participants were screened to see if they were over the age of 60; if so, the research was continued. The sample skewed to the younger cohort (i.e. 60-69, 70-79)
- Line 192 - Can the authors define “grassroots” and “middle-class?”
A phrase grassroots/lowest has been adopted to replace the word, grassroots, to illustrate for readers that these terms in the measurement are for reflecting the subjective views of the respondents in comparison with their view of the general public.
- Line 201-202 – 70.5% of individuals without any digital devices were female. What was the age distribution of those without any digital devices? Were females more likely to be older? Could this be an age issue as opposed to a gendered issue?
183 people said they did not use any digital devices. 31 participants in the 60-69 age group indicated they did not use any digital devices. 31 participants in the 60-69 age group indicated they did not use any digital devices. 65 participants in the 70–79 age group indicated they did not use any digital devices. 65 participants in the 80-89 age group indicated they did not use any digital devices. 22 people aged over 90 reported not having any digital devices.·
Line 225 – Was age also controlled for within the hierarchical multiple regression? If not, why? Again, “younger older adults” may be more likely to use technology than “the oldest older adults.” I can see age being an important predictor.
We have taken this into account and have re-run the regression analyses after controlling for age (along with other demographic variables). The results are basically the same. digital skills significantly predicted self-rated mental health (β = .232, p = .003) even after controlling for gender, education level, perceived social class, and age.
Limitation:
- One additional limitation which should be included is that only those individuals with access to telephones were included in the study, which excluded the most vulnerable older adults who do not have access to either a landline or smart phone.
Thanks for the advice. The suggested limitation has been included in the discussion:
… Second, only those with access to telephones have been included in the study. There is the possibility that the most vulnerable older adults who do not have access to landline or smart phones could be excluded.

Round 2
Reviewer 2 Report
The authors tried to respond to all issues addressed, and the quality and clarity of the paper considerably improved and made it worth publishing.